# The Bose polaron as thermometer of a trapped Bose gas: a quantum Monte Carlo study

Tomasz Wasak[1*], Gerard Pascual[2], Gregory E. Astrakharchik[2], Jordi Boronat[2] and Antonio Negretti[3]

**1** Institute of Physics, Faculty of Physics, Astronomy and Informatics, Nicolaus Copernicus University in Toruń, Grudziądzka 5, 87-100 Toruń, Poland
**2** Departament de Física i Enginyeria Nuclear, Universitat Politècnica de Catalunya, Campus Nord B4-B5, E-08034, Barcelona, Spain
**3** Zentrum für Optische Quantentechnologien, Fachbereich Physik, Luruper Chaussee 149, D-22761, Hamburg, Germany

* twasak@umk.pl

## Abstract

Quantum impurities interacting with quantum environments offer unique insights into many-body systems. Here, we explore the thermometric potential of a neutral impurity immersed in a harmonically trapped bosonic quantum gas below the Bose-Einstein condensation critical temperature $T_c$. Using ab-initio Path Integral Monte Carlo simulations at finite temperatures, we analyze the impurity's sensitivity to temperature changes by exploiting experimentally accessible observables such as its spatial distribution. Our results, covering a temperature range of $-1.1 \leqslant T/T_c \leqslant 0.9$, reveal that the impurity outperforms estimations based on a one-species bath at lower temperatures, achieving relative precision of 3-4% for 1000 measurement repetitions. While non-zero boson-impurity interaction strength $g_{BI}$ slightly reduces the accuracy, the impurity's performance remains robust, especially in the low-temperature regime $T/T_c \lesssim 0.45$ withing the analyzed interaction strengths $0 \leqslant g_{BI}/g \leqslant 5$, where $g$ is the boson-boson coupling. We confirm that quantum optical models can capture rather well the dependence of the temperature sensor on the impurity-gas interaction. Although our findings are in qualitative agreement with previous studies, our Monte Carlo simulations offer improved precision. We find that the maximum likelihood estimation protocol approaches the precision comparable to the limit set by the Quantum Fisher Information bound. Finally, using the Hellinger distance method, we directly extract the Fisher information and find that, by exploiting the extremum order statistics, impurities far from the trap center are more sensitive to thermal effects than those close to the trap center.

# 1   Introduction

Impurity physics with ultracold atomic quantum systems is a rather consolidated research branch, particularly for neutral impurities. Bose or Fermi impurities, commonly referenced to as *polarons*, have been extensively studied in bulk systems [1–4], while experimentally their properties have been measured and, recently, it has been possible to track the dynamics of such quasi-particles [5–13]. Albeit the physics of impurities has a long history – almost a century since the seminal works by Landau [14] and Pekar [15] – it is important to note that such a phenomena has been studied intensively in the laboratory only in the last decade, not only to confirm theoretical predictions, but also to gain a more comprehensive understanding of their properties, formation and, more intriguingly, how (quantum) correlations in such systems are generated.

It is interesting to note, however, that beyond quantum simulation purposes, such as polaron physics and mediated interactions, investigations into exploiting impurities for technological applications, such as sensing devices, have only recently been undertaken. Indeed, while the understanding of how polarons form and interact is key in our understanding of high-temperature superconductivity [16–18], sensing the environment's temperature of a quantum gas, utilized as a quantum simulator, is crucial for attaining the required temperature regime to observe, for example, mediated interactions in a Fermi gas [19]. Hence, temperature control is an essential tool for the calibration of quantum simulators. In this regard, proposals for measuring the temperature of a fermionic quantum gas with a neutral and charged impurity have already been put forward [20–22] as well as by using Bose polarons [23], where sub-nK thermometry has been predicted. Most importantly, those approaches are non-destructive, in contrast to, e.g., time-of-flight measurements, and can be even more precise. Moreover, probing temperature locally is imperative when energy transport in separated baths or thermalization after a quench is investigated [24, 25].

In a recent study, we have shown that the level of miscibility of impurities in a bosonic bath can be controlled by the bath's temperature. With the aid of quantum Monte-Carlo techniques it has been possible to accurately observe the transition from the miscible to immiscible regimes. An important observable is the mean position of the impurity, especially its relation to the gas temperature. Thus, on the basis of that study, in this work we investigate the thermometry features of such compound system. Based on ab-initio many-body simulations we perform a quantitative analysis going beyond quantum optical models such as that one developed in the proposal of Ref. [23]. While we confirm the good quality of the temperature sensitivity of

the Bose polaron, quantitatively we find that the actual thermometric performance is better when the Monte Carlo results are exploited. The difference relies essentially on the fact that in our approach we can take into account correlations, and, therefore, reveal the role of those in the sensitivity of the thermometer. Remarkably, we confirm the observation from Ref. [23] that the thermometric sensitivity weakly depends on the boson-impurity interaction strength.

The paper is organized as follows. In Sec. 2 we describe the compound system and provide the technical information regarding the many-body simulations, in Sec. 3 we provide the general framework of the employed parameter estimation theory approach. In Sec. 4 we analyze the thermometric performance exploiting simple observables such as the $k$-th moment of the impurity's position. In Sec. 5 we provide the temperature sensitivity by employing the maximum likelihood estimation procedure. Secs. 6 and 7 discuss results derived using the Hellinger distance method and order statistics, respectively. Finally, in Sec. 8 we summarize and draw our conclusions.

## 2 Bose polaron Hamiltonian

The Hamiltonian of the system containing $N$ bosons and a single impurity of equal mass, held in a harmonic trap in three dimensions, is

$$\hat{H} = \hat{H}_I + \hat{H}_B + \hat{H}_{IB} , \tag{1a}$$

with $\tag{1b}$

$$\hat{H}_B = -\frac{\hbar^2}{2m} \sum_{i=1}^{N} \nabla_i^2 + \sum_{n=1}^{N} V_{\text{ext}}(\mathbf{x}_i) + \sum_{i<j}^{N} V_{BB}(\mathbf{x}_{ij}) , \tag{1c}$$

$$\hat{H}_I = -\frac{\hbar^2}{2m} \nabla_I^2 + V_{\text{ext}}(\mathbf{x}_I) , \tag{1d}$$

and $\tag{1e}$

$$\hat{H}_{IB} = \sum_{i=1}^{N} V_{BI}(\mathbf{x}_{iI}) . \tag{1f}$$

Bose particle positions are $\mathbf{x}_i$, with $i = 1, \ldots, N$, and the impurity position is $\mathbf{x}_I$. The relative positions are denoted as $\mathbf{x}_{\alpha\beta} = \mathbf{x}_\alpha - \mathbf{x}_\beta$, where $\alpha, \beta \in \{i, I\}$. The first terms in Eq. (1c) and in Eq. (1d) describe the kinetic energies of the atoms and the impurity, respectively. The last term in Eq. (1c) and the Hamiltonian in Eq. (1f) are the boson-boson and boson-impurity interactions, respectively. Finally, the second terms in Eq. (1c) and Eq. (1d) describe the external trapping potentials of the particles.

For densities $n$, such that $na^3 \ll 1$ holds, the detailed form of the interatomic potential becomes unimportant and the interaction among the bosons as well as the impurity-boson one can be expressed only in terms of the $s$-wave scattering lengths [26]. In other words, the actual interaction potential can be replaced by a pseudopotential, whose coupling constant is given by $g = 4\pi\hbar^2 a/m$ for the boson-boson and $g_{BI} = 4\pi\hbar^2 a_{BI}/m$ for the boson-impurity interaction. The use of pseudopotentials is not possible in quantum Monte Carlo but, universality in terms of $a$, allows for using more convenient models [27]. For completeness, we remark here that we model the repulsive boson-boson interaction by $V_{BB}(r) = V_0/r^{12}$ and the boson-impurity by $V_{BI}(r) = V_0^{BI}/r^{12}$ with amplitudes $V_0$ and $V_0^{BI}$ adjusted to reproduce the three-dimensional $s$-wave scattering length $a$ and $a_{BI}$, respectively, as outlined in Refs. [28,29]. The external trap is assumed to be harmonic, i.e., $V_{\text{ext}}(r) = m\omega^2 r^2/2$, with the characteristic oscillator length scale $a_{\text{ho}} = \sqrt{\hbar/(m\omega)}$ and $\omega$ is the trap frequency.

In our numerical simulations we have at the trap center $na^3 < 10^{-5}$, whereas $a_{\text{ho}}/a = 15$

and the number of bosons $N \sim 100$. In our subsequent analyses the temperature scale is referred to the degeneracy temperature of an ideal BEC in a harmonic trap, namely

$$T_c = \omega \left( \frac{N}{\zeta(3)} \right)^{1/3}, \tag{2}$$

where $\zeta(x)$ is the Riemann zeta function, and hereafter we set $k_B = 1$ and $\hbar = 1$.

Similarly to Ref. [27], we employ the path integral Monte Carlo (PIMC) method for simulating $N$ bosons and the polaron at a temperature $T$ in a 3D harmonic trap (1). Briefly, we apply the *trotterization* algorithm [30] to the decoupled parts of the kinetic and potential operators of the thermal density matrix [31–34]

$$\hat{\rho}(T) = \frac{1}{Z} e^{-\hat{H}/T}, \tag{3}$$

where $Z = \text{Tr}\left[\exp(-\hat{H}/T)\right]$ is the partition function. The particle indistinguishability is imposed by sampling permutations using the worm algorithm [35].

With PIMC we are able to evaluate the one-particle distribution function of bosons $n_B(\mathbf{r}|T)$ as well as of the one of the impurity $n_I(\mathbf{r}|T)$ for fixed values of the temperature $T$ of the compound system at thermodynamic equilibrium, where $\mathbf{r}$ is the position of the particle. With the normalization $\int d^3 r \, n_\alpha(\mathbf{r}|T) = N_\alpha$, with $N_B = N$ and $N_I = 1$, we define the normalized probability distribution as

$$p_\alpha(\mathbf{r}|T) \equiv \frac{n_\alpha(\mathbf{r}|T)}{N_\alpha}, \tag{4}$$

where $\alpha = B$ for bosons and $\alpha = I$ for the impurity. Hence, in our parameter estimation analyses, we employ these functions obtained from PIMC simulations of the compound system to determine the ultimate precision of estimating the temperature of the bath from observables. As an example of observable, we take the position of the impurity relative to the trap center.

## 3 Thermometric performance

For the sake of clarity, we briefly review the main concepts of estimation theory that are relevant for the present work. We note that a rather detailed discussion is also provided in Sec. V of Ref. [21].

Our objective is to measure the gas temperature non-destructively. Hence, we focus on the measurable properties of the impurity, namely its position in the trapped gas. The measurement (Hermitian) operator, $\hat{\Pi}_I(\mathbf{r}) = |\mathbf{r}\rangle\langle\mathbf{r}|$, of the impurity's position observable leads to a probability distribution for the measurement results $\mathbf{r}$ given by $p_I(\mathbf{r}|T) = \text{Tr}\left[\hat{\Pi}_I(\mathbf{r})\hat{\rho}(T)\right]$, where the atoms-impurity density matrix is given in Eq. (3). The distribution is already normalized accordingly to the completeness relation $\int d^3 r \, \hat{\Pi}(\mathbf{r}) = 1$. Towards a less cumbersome notation, here we drop the index $I$ on the impurity's measurement results of the chosen observable.

After performing $M$ independent measurements, the temperature is estimated from the estimator $T_{\text{est}}(\mathbf{r}_1, \dots, \mathbf{r}_M)$, which is a scalar-valued function of the measured outcomes distributed according to $P_I^{(M)}(\mathbf{r}_1, \dots, \mathbf{r}_M|T) = \prod_{i=1}^M p_I(\mathbf{r}_i|T)$. The inferred temperature, $T_{\text{est}}$, has the property that $\langle T_{\text{est}} \rangle = T$, where the statistical average is taken over $P_I^{(M)}$. Importantly, this implies that the estimator is unbiased. The uncertainty of the estimation, which is due to both inherent quantum randomness and classical lack of knowledge of microscopic states, deteriorates thermometric performance, thus manifesting in a statistical uncertainty of the estimate $T_{\text{est}}$. The variance of $T_{\text{est}}$, denoted with $\Delta^2 T_{\text{est}}$, determines the uncertainty of the temperature

estimation and, thus, the thermometric performance. The so-called Cramer-Rao lower bound (CRLB) [36] determines the minimum estimation sensitivity, and it is given by

$$\Delta^2 T_{\text{est}} \geqslant \frac{1}{MF_T}, \tag{5}$$

where the classical Fisher information (CFI) is given by:

$$F_T \equiv \int d^3r \, p_I(\mathbf{r}|T)(\partial_T \ln p_I(\mathbf{r}|T))^2. \tag{6}$$

$F_T$ is the central figure of merit for quantifying thermometric performance and can be interpreted as the amount of information about $T$ contained in the distribution $p_I(\mathbf{r}|T)$. An estimator, which saturates asymptotically, i.e., in the limit $M \to \infty$, the CRLB is the so-called Maximum Likelihood Estimator (MLE) [36].

If measurement data are employed to construct average values of observables $\langle \hat{A} \rangle_T$, which depend on the unknown temperature $T$, and if the relation for various $T$ is known because it was determined either numerically, analytically or experimentally, the temperature can be inferred from the average value by inverting the functional form of $\langle \hat{A} \rangle_T$ as a function of $T$. Thus, in such a protocol the uncertainty is set by

$$\Delta^2 T_{\text{est}} = \frac{\Delta^2 \hat{A}}{M \chi_T^2(\hat{A})}, \tag{7}$$

where $\chi_T(\hat{A}) \equiv \text{Tr}[\partial_\xi \rho(\xi)\hat{A}]|_{\xi=T} = \partial_T \langle \hat{A} \rangle_T = \partial A(T)/\partial T$ represent the static susceptibility of the observable $\hat{A}$ and $A(T) = \langle \hat{A} \rangle_T$ is the equilibrium average at temperature $T$. In this protocol, that is in the estimation from average values, the full probability distribution is not exploited, so some information about $T$ is inevitably lost, and, in general, the uncertainty from Eq. (7) is larger that the bound from Eq. (5), i.e.,

$$\frac{\chi_T^2(\hat{A})}{\Delta^2 \hat{A}} \leqslant F_T. \tag{8}$$

This relation helps to determine the thermometric performance when the access to the distribution function of the outcomes is challenging and only average values or their fluctuations can be determined.

Above we have focused on the measurement of the position of the impurity, i.e., $\hat{\Pi}_I(\mathbf{r})$. Quantum mechanics, however, allows for the choice of various measurements that are given by positive semi-definite operators $\hat{E}_i$ that sum up to the unit operator $\sum_i \hat{E}_i = 1$, and the index $i$ labels the outcomes of the measurement represented by the set of operators $\hat{E}_i$. Admitting that the CFI depends on the measurement, i.e., $F_T(\hat{E}_i)$, optimization with respect to the operators $\hat{E}_i$ implies

$$F_T \leqslant \mathcal{F}_T, \tag{9}$$

where the quantum Fisher information (QFI) is given by $\mathcal{F}_T = \max_{\{\hat{E}_i\}} F_T(\hat{E}_i)$. The value of the QFI can be also expressed as $\mathcal{F}_T = \text{Tr}[\hat{\rho}(T)\hat{\Lambda}_T^2]$ with the implicitly defined Symmetric Logarithmic Derivative (SLD) $\hat{\Lambda}_T$ given by $\partial_T \hat{\rho}(T) = [\hat{\rho}(T)\hat{\Lambda}_T + \hat{\Lambda}_T \hat{\rho}(T)]/2$. Since the thermal state is diagonal in the energy representation, the SLD can be even determined analytically. The final result is given by the variance of the Hamiltonian in the thermal state, namely

$$\mathcal{F}_T = \frac{\Delta^2 \hat{H}}{T^4}. \tag{10}$$

For reference, we note that for a single thermalized impurity held in a 3D harmonic trap, the QFI can be expressed [37] as

$$\mathcal{F}_T^{\text{non}} = \frac{1}{T^2} \frac{3(\beta\omega/2)^2}{\sinh^2(\beta\omega/2)}, \tag{11}$$

where the superscript "non" refers to the case of the impurity not interacting with the bosonic gas and $\beta = 1/T$ (remind that $k_B \equiv 1$).

To see how the presence of the impurity improves the thermometric performance, one can investigate the QFI in the presence of the impurity and without, i.e.,

$$\zeta_T = \mathcal{F}_T[\hat{\rho}_{BI}] - \mathcal{F}_T[\hat{\rho}_B], \tag{12}$$

where we explicitly denoted the dependence on the thermal state of the atom-impurity system $\hat{\rho}_{BI} \propto \exp(-\beta\hat{H})$, with $\hat{H}$ from Eq. (1). Here, $\hat{\rho}_B \propto \exp(-\beta\hat{H}_B)$ and $\hat{H}_B$ is the Hamiltonian $\hat{H}$, but with all the terms containing the impurity operators removed. If $\zeta_T < 0$, in principle there is no gain in employing the impurity for enhanced thermometric performance, as its presence deteriorates the temperature sensitivity as well.

Exploiting the impurity limit, we assume that in the first approximation, the density matrices do not change by the presence of the impurity in terms that involve averages of bosons only. Consequently, $\zeta_T$ takes the form

$$\begin{aligned}
T^4\zeta_T \approx & \langle\hat{H}_I^2\rangle - \langle\hat{H}_I\rangle^2 \\
& + \langle\hat{H}_{IB}^2\rangle - \langle\hat{H}_{IB}\rangle^2 \\
& + \langle\hat{H}_B\hat{H}_{IB} + \hat{H}_{IB}\hat{H}_B\rangle - 2\langle\hat{H}_B\rangle\langle\hat{H}_{IB}\rangle \\
& + \langle\hat{H}_I\hat{H}_{IB} + \hat{H}_{IB}\hat{H}_I\rangle - 2\langle\hat{H}_I\rangle\langle\hat{H}_{IB}\rangle.
\end{aligned} \tag{13}$$

Here, all the averages are calculated with the full equilibrium density matrix from Eq. (3) describing $N$ atoms and 1 impurity. In deriving the approximation we dropped the terms $\langle H_B^2\rangle - \langle H_B^2\rangle_N$ and $\langle H_B\rangle^2 - \langle H_B\rangle_N^2$, where the average with subscript $N$ is taken over the state of bosons only $\propto \exp(-H_B/T)$. The first term in Eq. (13) represents the equilibrium contribution of the impurity alone. Importantly, this term is given by the average over the state $\hat{\rho}_I(T) \equiv \text{Tr}_N\hat{\rho}(T)$, where the trace is taken over $N$ atoms, which can be far from equilibrium distribution $\propto \exp(-\hat{H}_I/T)$ especially in the strong-coupling when $g_{IB}/g$ is large. The second (and positive) term contains the contribution owed to the impurity-bath interaction, whereas the last two terms represent correlations that can be either positive or negative. The latter, however, is crucial for the final thermometric performance of the impurity contribution.

The calculations of the fluctuations that appear in Eq. (13) within the PIMC method are very challenging, and are beyond the scope of this study. Instead, in the following, we consider the thermometric performance of the Bose impurity in the case when the temperature is estimated either from the averages of the $k$-th moment of the impurity position or from its full distribution, i.e., Eq. (4). Such an approach directly probes the density matrix of $\hat{\rho}_I(T)$ of the impurity and its temperature dependence.

## 4 Estimation from the average impurity position

The many-body simulations based on the PIMC provide us with the impurity density $n_I(\mathbf{r}_n|T_j)$ on the grid $\mathbf{r}_n$, where $n$ is the index of the grid point, and $T_j$ is $j$-th point of the grid in the temperature used in our numerics. Since the trap frequencies are equal in all the spatial directions, $n_I$ is spherically symmetric, and we obtain a function that depends on the distance $|\mathbf{r}_n|$; the grid points in the space $|\mathbf{r}|$ are denoted with $r_i$ for the $i$-th point.

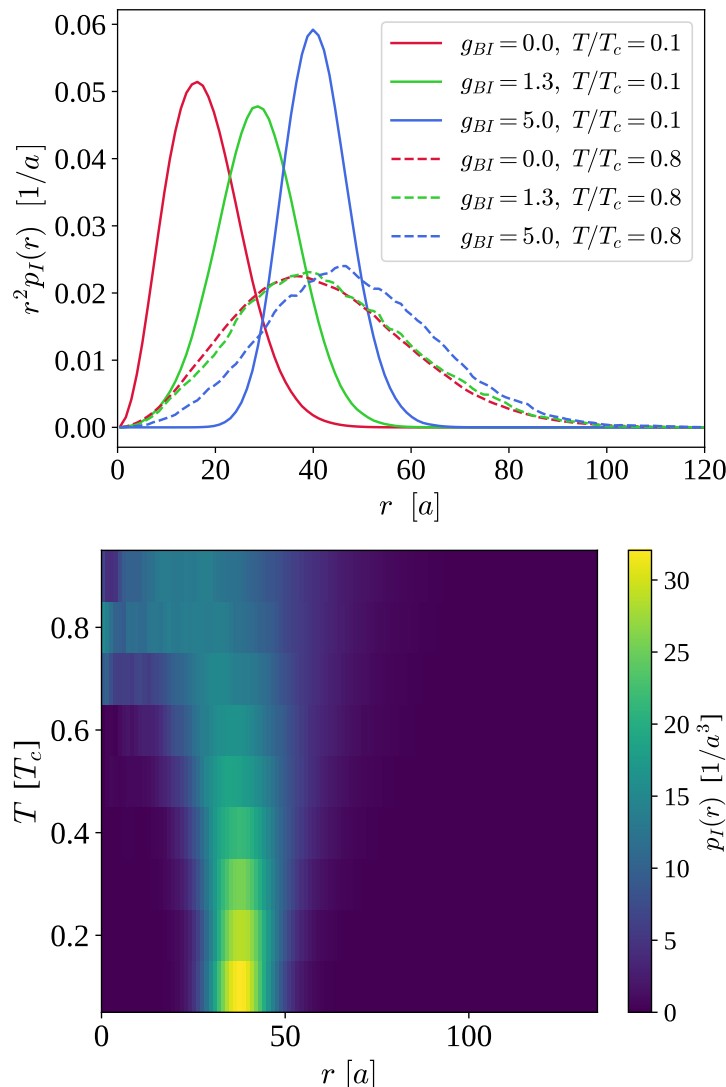

Figure 1: Top panel: Impurity probability density $r^2 p_I(r|T)$ for $T = 0.1$ (solid) and 0.8 (dashed) (in units of $T_c$) and for interaction strength $g_{BI} = 0$ (red), 1.3 (green), 5.0 (blue) (in the units of $g$). Bottom panel: the impurity probability density distribution $p_I(r|T)$ for $g_{BI}/g = 5.0$ as a function of the distance from the trap center $r$ and the temperature $T$ of the Bose gas.

For a fixed value of $T$, from the points $r_i$ we construct a spline-based interpolation that yields a continuous function $p_I(r|T)$ from which we extract the information about the thermometric performance of the impurity, as shown below. The normalization is given by $\int_0^\infty r^2 p_I(r|T) dr = 1$.

In the upper panel of Fig. 1 we show the impurity probability distributions for $g_{BI}/g = 0$ (red), 1.3 (green), 5.0 (blue) for low $T/T_c = 0.1$ (solid) and high $T/T_c = 0.8$ (dashed) temperatures. Since the probability distributions are very different for low and high-temperature regimes, the impurity is sensitive to the temperature variations. Interestingly, for large temperatures the variation in $g_{BI}$ does not affect substantially the shape of the function, which will have an impact on the thermometric performance.

The simplest observable from which the temperature of the gas can be estimated is the

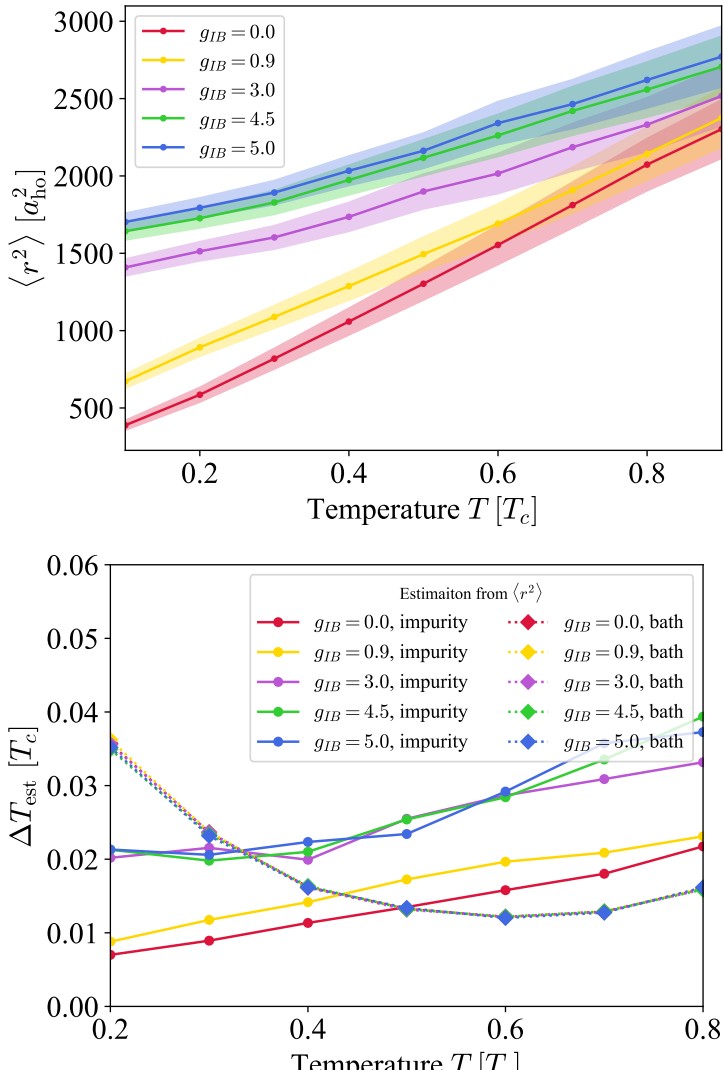

Figure 2: **(Top panel)** The impurity's mean-squared distance $\langle \mathbf{r}^2 \rangle$ from trap center as a function of temperature for $g_{IB} = 0, 0.9, 3.0, 4.5, 5.0$ (in the units of $g$). The shaded region corresponds to the scaled (by a factor of 10) standard deviation, i.e., the upper/lower limit is given by $\langle \mathbf{r}^2 \rangle \pm \sqrt{\langle \mathbf{r}^4 \rangle - \langle \mathbf{r}^2 \rangle^2}/10$. **(Bottom panel)** Uncertainty of the estimation (solid lines with circles) from the squared distance of the impurity (solid) $\langle \mathbf{r}_I^2 \rangle_T$ from the center of the harmonic trap for various values of $g_{BI}$ (in the units of $g$) and for $M = 1000$ statistical repetitions, see Eq. (7). For comparison, the estimation from the same observable for the Bose gas atoms is showed (dotted lines with diamonds).

mean $k$-th power of the position of the impurity:

$$\langle |\hat{\mathbf{r}}_I|^k \rangle_T \equiv \int d^3 r_I |\mathbf{r}_I|^k p_I(\mathbf{r}_I|T). \tag{14}$$

This choice corresponds in Eq. (7) to the observable $\hat{A} = |\hat{\mathbf{r}}_I|^k$. Consequently, to assess the corresponding uncertainty of estimation, we need the fluctuations $\Delta^2 \hat{A} = \langle |\hat{\mathbf{r}}_I|^{2k} \rangle_T - \langle |\hat{\mathbf{r}}_I|^k \rangle_T^2$. To evaluate the static susceptibility $\chi_T(|\hat{\mathbf{r}}_I|^k)$, however, we evaluate the function $\langle |\hat{\mathbf{r}}_I|^k \rangle_T$ on the available grid of $T$. Using the spline interpolation method, as we mentioned before, we can evaluate the derivative, that is, $\chi_T = \partial_T \langle |\hat{\mathbf{r}}_I|^k \rangle_T$.

In Fig. 2 (top panel), we present the impurity's mean-squared distance $\langle \mathbf{r}^2 \rangle$ from trap center as a function of temperature for various values of $g_{IB}/g$. In addition, we display, as a shaded region, the corresponding standard deviation (scaled by a factor of 10 for visibility) of the observable $\hat{A} = \hat{\mathbf{r}}^2$; the scaling is just for presentation clarity. Along with increasing the interaction strength the slope (see Eq. (7)) of the function decreases, while the standard deviation remains of the same order. This results in degraded sensitivity of the impurity as shown in bottom panel, i.e., the impurity uncertainty $\Delta T_{\text{est}}$ increases with increasing interaction strength $g_{IB}/g$. For low temperatures and large interactions strenghts, the impurity is repelled to the edges of the atomic cloud and therefore, the dependence on the temperature of its position becomes weak.

In Fig. 2 (bottom panel), we show the resulting uncertainty of the estimation $\Delta T$, in the units of $T_c$ for various values of the coupling constant $g_{BI}/g$ for the choice $\hat{A} = \hat{\mathbf{r}}_I^2$, i.e., $k = 2$ in Eq. (14). For comparison, we provide the uncertainty of the estimation in the case when the position of a single atom from the cloud is measured. To evaluate it, we follow the same procedure as described above, but in Eq. (14) we have replaced the distribution $p_I$ with $p_B$ defined as in Eq. (4).

From the results for the impurity, we find that for $M = 1000$ repetitions the uncertainty is, for weak coupling, on the order of $\sim 0.01 T_c$ for low temperatures of the gas, and $\sim 0.02 T_c$ for the high temperature regime. In the strong coupling regime, for low-$T$ the uncertainty is $\Delta T_{\text{est}} \sim 0.02 T_c$ and for high-$T$ it is $\Delta T_{\text{est}} \sim 0.04 T_c$. Interestingly, the dependence on the strength $g_{BI}/g$ is weak in the whole temperature range (gain of a factor of 2 in the weak coupling compared to strong coupling). This can be understood from Fig. 1 (upper panel), since, in terms of the distribution $r^2 p_I(r|T)$ which enters the uncertainty $\Delta T$ in Eq. (7), the effect of increasing the interaction $g_{BI}$ is to shift the distribution peak to larger $r$ and alter its width. For instance, as seen from the figure by comparing $T/T_c = 0.1$ and $T/T_c = 0.8$, the position of the peak for $g_{BI}/g = 1.3$ (green) has larger derivative with respect to $T$ a compared to $g_{BI}/g = 5.0$ (blue), but the width of the distribution for $g_{BI}/g = 1.3$ is larger, which partially compensates the effect of the larger derivative of the peak position. In the low temperature regime, however, the interaction with the bath deteriorates the thermometric performance. Inspecting Fig. 1 (upper panel), shows that the widths of distributions for $g_{BI}/g = 0$ and $1.3$ are similar, but the peak position for nonzero interaction is far from the trap center. Since the high-$T$ distributions are similar, the estimation from $g_{BI} = 0$ case is more susceptible to temperature change yielding lower uncertainty. Notwithstanding, the impurity thermometry is still better by a factor of $\sim 2 - 3.5$ than the estimation from $\langle \mathbf{r}^2 \rangle$ of the atoms from the Bose gas (see Fig. 2 as well, dotted lines with diamonds) in the low temperature regime. For higher temperatures, the estimation from the gas is slightly more beneficial for thermometry.

In Fig. 3, we show the estimation from the impurity fourth moment of the distance from the trap center, i.e., $\hat{A} = \hat{\mathbf{r}}_I^4$, so with $k = 4$ in Eq. (14). Since this observable is more sensitive to the shape of the distribution far from the trap-center and also to its tails, the overall thermometric performance increases. Similarly to the previous estimation protocol, we observe here that the impurity has better thermometric performance for small temperatures by a factor of 2-3 compared to estimation from bath atoms. Interestingly, we find that for weak interactions, the estimation from $\langle \mathbf{r}^2 \rangle$ and $\langle \mathbf{r}^4 \rangle$ provide similar thermometric performance. However, it is slightly better to estimate temperature from $\langle \mathbf{r}^4 \rangle$ in the strong coupling regime indicating that more information about the temperature is encoded in large distances $|\mathbf{r}|$ from the trap center.

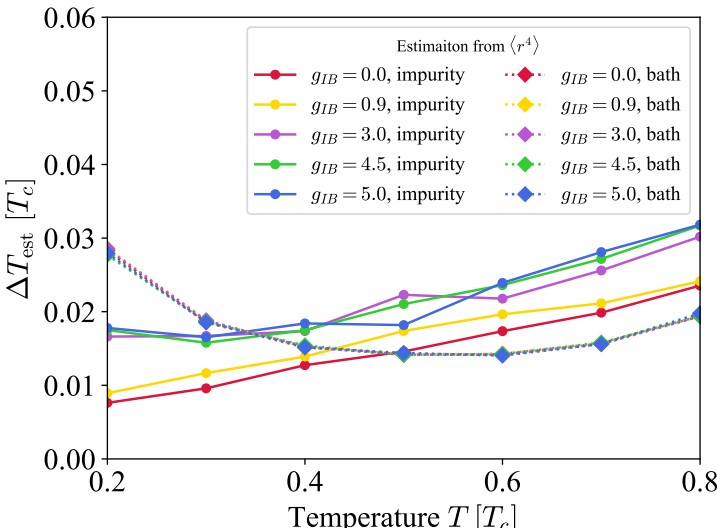

Figure 3: Uncertainty of the estimation from the impurity (solid with dots) distance $\langle \mathbf{r}_I^4 \rangle_T$ for various values of $g_{12}$ (in the units of $g$) and for $M = 1000$ statistical repetitions, see Eq. (7). The estimation from $\langle \mathbf{r}^4 \rangle_T$ of the Bose gas atoms is showed for comparison (dotted lines with diamonds).

## 5  Estimation from the distribution

Towards the assessment of the Fisher information from Eq. (6), we need first to determine the derivative of the distribution function. However, since it is known only on a discrete set of $T$, its evaluation is numerically noisy. Therefore, we resort to the MLE estimation, which for large sample size $M$ asymptotically saturates the bound in Eq. (5).

Specifically, we construct the likelihood function $L_M(\tilde{T})$ which is given by $L_M(\tilde{T}) \equiv \sum_{i=1}^{M} \ln(p_I(r_i|\tilde{T}))$, where the samples $r_i$ are generated from the distribution $p_I(r|T)$ with the actual temperature $T$. We note here, that $\tilde{T}$ here is merely a parameter and $T$ is unknown for the estimation protocol, i.e., only the set of outcomes $\{r_i\}$ is available for the protocol. Then, we construct the estimator $T_{\mathrm{est}} \equiv \arg\max_{\tilde{T}} L_M(\tilde{T})$, i.e., we search for $\tilde{T}$ which yields the maximum of the likelihood function $L_M(\tilde{T})$. $T_{\mathrm{est}}$ is a random variable because it depends on the random sample $\{r_i\}$, and it is the estimate of the actual temperature $T$. In order to assess the uncertainty of the estimation, we repeat the procedure described above $M_{\mathrm{MLE}}$ times, and determine the variance of $T_{\mathrm{est}}$. In order to work with continuous distribution, we perform a spline fit to the two-dimensional function $Q(r, T)$ in the following form $p_I(r|T) = e^{-Q(r,T)}$, which ensures that $p_I \geqslant 0$ and shows better numerical stability.

In Fig. 4 we present the results for the inferred uncertainty $\Delta T$ for the MLE estimation for the number of measurements $M = 1000$ and $M_{\mathrm{MLE}} = 200$ repetitions of the procedure. Comparison to the uncertainty given by the Cramer-Rao lower bound, as extracted from the Hellinger distance in Sec. 6 below, shows that the MLE results for these sample sizes are in good agreement with the bound. Here, for low-$T$ regime we get uncertainty on the order $\Delta T \sim 0.010 - 0.015 \, T_c$, and for $T \sim 0.5T_c$, $\Delta T \sim 0.012 - 0.015 \, T_c$. In terms of the relative uncertainty, $\Delta T_{\mathrm{est}}/T \sim 5-8\%$ in the low temperature regime $T = 0.2T_c$ and $\Delta T_{\mathrm{est}}/T \sim 3.5\%$ for $T \gtrsim 0.5T_c$.

As expected, the results yield slightly better performance as compared to the estimation from the mean values of observables, which indicates that the estimation from the $k = 2$ moment $\langle |\mathbf{r}|^2 \rangle$ contains already much information about the temperature. However, we notice, similarly to the estimation from the results from Fig. 2 or Fig. 3, the uncertainty weakly

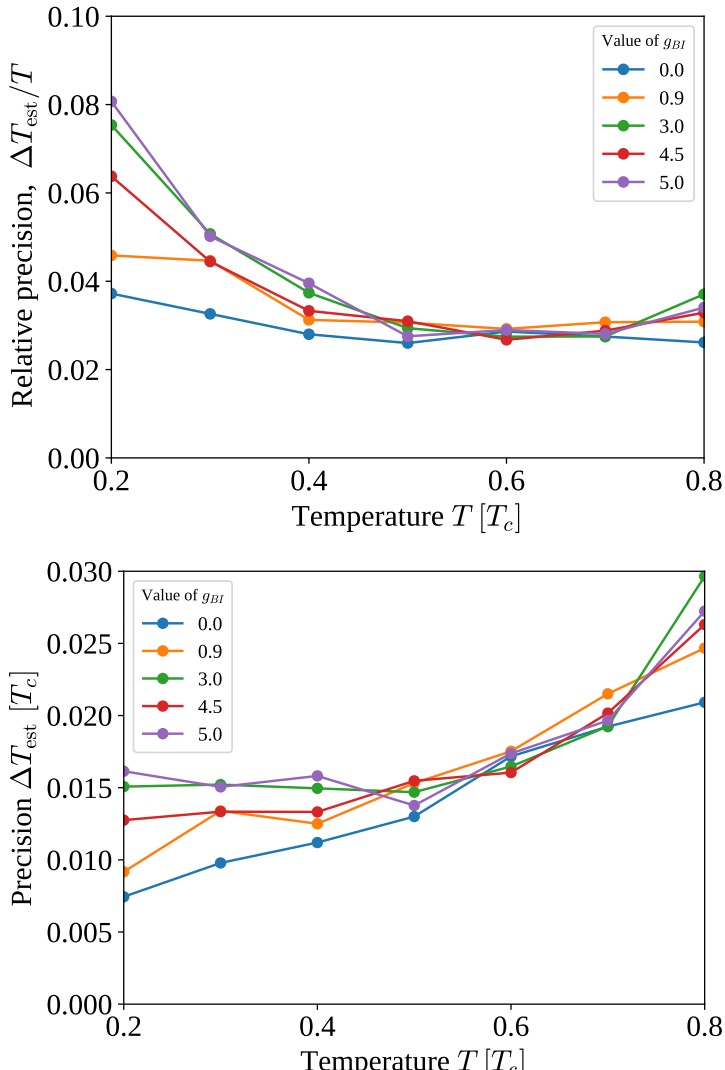

Figure 4: Relative $\Delta T/T$ (top panel) and absolute $\Delta T$ (bottom panel) uncertainty of the temperature for $g_{BI} = 0$, 0.9, 4.5 5.0 (in units of $g$) as a function of the temperature (in units of $T_c$). The uncertainty is given by the Maximum Likelihood Estimation protocol with $M = 1000$ samples and $M_{\mathrm{MLE}} = 200$ estimation repetitions to evaluate the uncertainty.

depends on the interaction strength $g_{BI}$ of the impurity with the trapped gas. We find that the relative thermometric precision is on the order of $3 - 6\%$ across the whole investigated temperature range for the selected number of measurements.

# 6 Hellinger distance method

The estimation from the full probability distribution involved drawing samples which were subsequently used in maximum likelihood estimation protocol. Here, we exploit the squared Hellinger distance method, that was developed in Ref. [38], in order to extract the value of the Fisher classical information.

In our context, the Hellinger distance $d_H(T|T')$ is defined by

$$d_H^2(T|T') = \frac{1}{2} \int d^3r \left[ \sqrt{p_I(\mathbf{r}|T)} - \sqrt{p_I(\mathbf{r}|T')} \right]^2, \tag{15}$$

and it quantifies the distance between the probability distributions $p_I(\mathbf{r}|T)$ and $p_I(\mathbf{r}|T')$ with two different temperatures $T$ and $T'$, respectively. Rewriting now, $T' = T + \delta T$, we arrive at, up to second order in $\delta T$

$$d_H^2(T|T') = \frac{F_T}{8} \delta T^2 + \mathcal{O}(\delta T^3). \tag{16}$$

This formula relates the CFI and the Hellinger distance, showing that the curvature of $d_H^2$ as a function of $\delta T$ yields $F_T$. Therefore, we exploit this connection by numerically evaluating $d_H^2(T|T + \delta T)$ for a given $T$ and scanning over available values of $\delta T$. Then, a quadratic fit yields $F_T$.

In Fig. 5, we provide the results of determining $F_T$ within the fitting method for the interaction strength $g_{BI}/g = 0$, 0.9, 4.5 5.0 and as a function of temperature $T/T_c$. The dots correspond to the CFI extraction from the impurity distribution function $p_I(\mathbf{r}|T)$, while diamonds correspond to the evaluation of the CFI from the Bose-gas atom distribution $p_B(\mathbf{r}|T)$. We observe a similar behavior of $F_T$ as in Fig. 4. Namely, in the low-$T$ regime, the best thermometric performance is for weak couplings, and the sensitivity deteriorates with increased interaction strength. We also observe in Fig. 5, but now more pronounced, the independence of the sensitivity with the coupling strength for large $g_{BI}/g$. Finally, for larger $g_{BI}/g$, the dependence of $F_T$ on $T$ is rather weak. The sensitivity evaluated from the Bose-gas atom distribution (dotted lines with diamonds) exhibits behavior similar to the data presented in Fig. 2 and Fig. 3. Specifically, the dependence on the interaction strength is negligible, while the uncertainty increases as the temperature decreases. In the low-temperature regime, impurity thermometry surpasses the ultimate precision of atomic thermometry, as defined by the Cramér-Rao bound of the Bose-gas atom distribution.

## 7 Order statistics

In this section we indicate which part of the impurity distribution, shown for example in Fig. 1 (top panel), contains more information about the temperature dependence. Specifically, we ask to which extent the position distribution of the impurity for small $|\mathbf{r}| \approx 0$ (center of the trap) or for large $|\mathbf{r}| \gg a_{\text{ho}}$ (trap edges) are sensitive to temperature of the Bose gas. To this end, we analyze the largest and smallest order statistics [39] of sample size $m$ taken from $M$ measured outcomes. Specifically, we divide $M$ measured outcomes $r_1, \ldots, r_M$, where each $r_i = |\mathbf{r}_i|$ is the measured distance from the trap center of the impurity position $\mathbf{r}_I$ in the $i$-th measurement outcome (drawn from the distribution $p_I(\mathbf{r}|T)$), into $m = M/n$ sets, each set of length $n$. If we take the largest value of the distance from each set, we obtain the largest order (order $k = n$) statistic, which probes large values of the the probability distribution of $|\mathbf{r}|$, i.e., its tails. On the other hand, if we take the minimum distance of each set, we obtain the smallest order (order $k = 1$) statistics, probing the $\mathbf{r} \approx 0$ region of the distribution $p_I(\mathbf{r}|T)$. Below, we exploit these extreme (smallest and largest) order statistics to analyze the thermometric performance of the impurity.

We note that when considering the order statistics, we obtain a smaller number, equal to $M/n$, of measurement outcomes that enter into the estimation protocol. This reduction can in principle reduce the sensitivity of the estimation, because part of the number of measurement outcomes are removed from considerations. This effect, however, is reduced by the fact that the probability distribution, upon which the estimation is based, changes, as we see below.

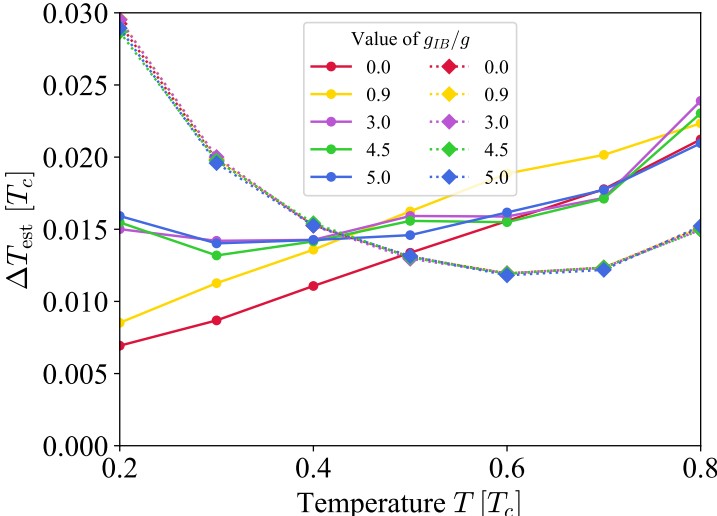

Figure 5: The uncertainty of the temperature as given by the Cramer-Rao lower bound, see Eq.(5), i.e., $\Delta T_{\text{est}} = 1/\sqrt{MF_T}$, where $F_T$ is extracted from the quadratic fit within the Hellinger distance method from the impurity distribution $p_I(\mathbf{r}|T)$ (solid lines with dots), see Eq. (16), for $g_{BI} = 0$, 0.9, 4.5 5.0 (in units of $g$) as a function of the temperature (in units of $T_c$). Dashed lines with diamonds correspond to uncertainty evaluated from the Bose gas atom distribution $p_B(\mathbf{r}|T)$. The uncertainty is calculated for $M = 1000$ samples.

The $k = 1$ and $k = n$ order statistics are described by the following probability distributions, respectively,

$$p_I^{(1,n)}(r|T) \equiv n\, p_{\text{r}}(r|T)[1 - F_I(r|T)]^{n-1}, \tag{17a}$$

$$p_I^{(n,n)}(r|T) \equiv n\, p_{\text{r}}(r|T)[F_I(r|T)]^{n-1}, \tag{17b}$$

where $p_{\text{r}}(r|T)$ is the radial distribution function of the distance $r = |\mathbf{r}|$, which is obtained from $p_I(\mathbf{r}|T)$ after integrating out the angles, i.e., it is given by $4\pi|\mathbf{r}|^2 p_I(\mathbf{r}|T)$, where we used the spherical symmetry of the system. The object $F_I(r|T)$ is the cumulative distribution function of the positions, i.e, $F_I(r|T) = \int_0^r dr' p_{\text{r}}(r'|T)$. We see that in Eqs. (17), the factor multiplying the distribution $p_{\text{r}}(r|T)$ can be interpreted as an envelope function that is either nonzero for small $r$, see Eq. (17a), or nonzero for large $r$, see Eq. (17b). We note here that one could investigate orders $1 < n < k$ of the order statistics, which are intermediate between the extremal ones. These would probe intermediate values of the distribution $p_{\text{r}}(r|T)$ by multiplying it with relevant envelope functions. Here, however, we focus only on the extreme orders of $k = 1$ and $k = n$.

The thermometric performance of the impurity within the protocol based on the order statistics is quantified by the value of the CFI corresponding to the distributions from Eqs. (17), namely,

$$F_T^{(k,n)} \equiv \int_0^\infty \frac{1}{p_I^{(k,n)}(r|T)} \left( \frac{\partial p_I^{(k,n)}(r|T)}{\partial T} \right)^2 dr. \tag{18}$$

In our numerical simulations, we extract the value of the CFI from Eq. (18) by the Hellinger distance method, as described in Sec. 6. Then, the precision of the estimation is determined by the Cramér-Rao lower bound and is expressed as $\Delta T_{\text{est}} = [F_T^{(k,n)} M/n]^{-1/2}$, where $M$ is the total number of outcomes, which is assumed to be divisible by the sample length $n$.

In Fig. 6, we present the uncertainty $\Delta T_{\text{est}}$ as a function of the temperature $T$ for $g_{BI}/g = 0.9$ (red lines) and $g_{BI}/g = 5.0$ (blue lines). The dots represent the estimation results obtained from the full distribution (see Eq. (16)), while the diamonds correspond to the smallest order statistic ($k = 1$; see Eq. (17a)), and the triangles depict the results for the largest order statistic ($k = 5$; see Eq. (17b)). In this analysis, the sample length is $n = 5$, and the total number of measurement outcomes is fixed at $M = 1000$.

The results demonstrate that the probability distribution $p_I(\mathbf{r}|T)$ only weakly depend on the temperature for $\mathbf{r} \approx 0$, since the uncertainty corresponding to the CFI for $k = 1$ order statistic (red/blue diamonds) is well above the uncertainty determined by the full probability distribution (red/blue circles) and also above the $k = 5$ order statistics (red/blue triangles) for each value of $g_{BI}/g$. Also, for each value of $g_{BI}/g$ the estimation from $k = 5$ order statistics (triangles) is relatively close to the full distribution (circles) and yield similar thermometric performance. This indicates that, indeed, much of the information about the temperature is contained in the distribution corresponding to large distances from the trap center.

Finally, we observe that the small-$r$ behavior of the distribution, which was investigated in Ref. [27], has impact on the temperature estimation efficiency from $k = 1$ order statistic (diamonds), as its dependence on $T$ exhibits qualitatively different trends for large and small $g_{BI}/g$. Namely, for large $g_{BI}/g$ the uncertainty $\Delta T_{\text{est}}$ (blue diamonds) is similar across the temperature range, as, in that interaction regime, the impurity is repelled to the edge of the Bose gas cloud and, thus, it becomes coupled to thermal wings of the Bose gas. On the other hand, the large interaction strength $g_{BI}/g = 5.0$ overshadows thermal effects, and, thus, in the low-temperature regime, the temperature uncertainty is larger as compared to the weaker interacting case $g_{BI}/g = 0.9$ (blue diamonds).

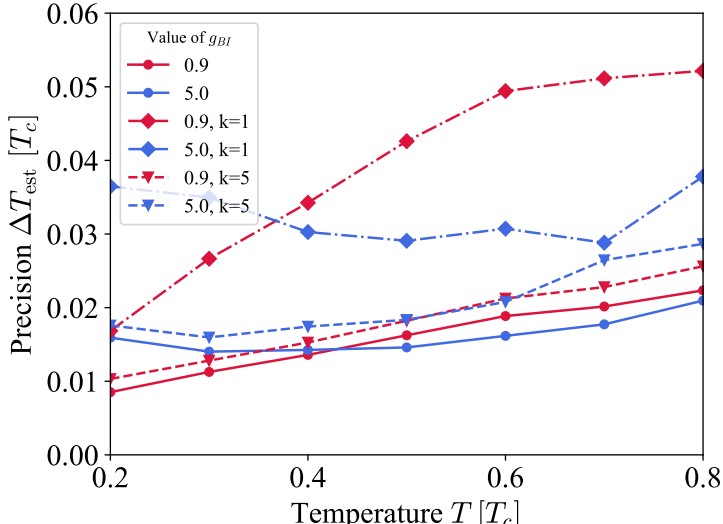

Figure 6: The uncertainty of the temperature $\Delta T_{\text{est}} = 1/\sqrt{F_T^{(k,n)} M/n}$, where $F_T^{(k,n)}$ is the CFI of the $k$-th order statistic of sample size $n$, extracted from the quadratic fit within the Hellinger distance methods, see Eq. (16), for $g_{BI} = 0.9$ (red) and 5.0 (blue) (in units of $g$) as a function of the temperature $T$ (in units of $T_c$). The shown orders are $k = 1$ (smallest order, diamonds) and 5 (largest order, triangles) for sample size $n = 5$. The uncertainty is calculated for $M = 1000$ samples.

# 8  Conclusion

In this work, we investigated the thermometric performance of an impurity interacting with a trapped gas of bosons below the Bose-Einstein condensation critical point $T_c$. We determined the distribution of positions of the impurity in the gas at finite temperatures employing the PIMC method.

Combining the Monte Carlo results with the methods of estimation theory, we analyzed the thermometric performance of the impurity playing the role of thermometer, when the temperature is inferred from the available observables. Specifically, we inferred the temperature of the bosonic gas in three cases: from the mean value of squared distance $\mathbf{r}^2$ of the impurity from the trap center, from the $k = 4$th moment, and from the whole distribution of impurity positions in the gas. In fact, the calculation of the Fisher information, which takes into account the whole distribution function, turned out to be numerically unstable due to numerical derivatives, and, therefore, we performed the Maximum Likelihood Estimation protocol in order to assess the sensitivity for $M = 1000$ repetitions of the measurements.

To validate the accuracy of the Maximum Likelihood Estimation protocol, we also employed a method for directly extracting the Fisher information based on the Hellinger distance, as developed in Ref. [38]. This approach also enabled an analysis of the extremum order statistics, providing insights into which parts of the position distribution function carry information about the temperature. Notably, at low temperatures, the smallest order statistics is predominantly influenced by interactions rather than thermal effects. In contrast, the largest order statistics captures information about the temperature of the Bose gas, demonstrating a performance comparable to that of the entire position distribution of the impurity across all interaction strengths.

By investigating the temperature range $0.1 \leqslant T/T_c \leqslant 0.9$, we found that the impurity's thermometric performance for a given measurement becomes comparable to that of the estimation performed on the bath atoms when $T/T_c \sim 0.45$. At lower temperatures, the impurity outperforms single-atom estimations of the bath particles as a thermometer, even when employing the same type of measurement.

We found that estimating the temperature from the fourth moment of the impurity's position distribution within the gas provides greater precision, $\Delta T_{\text{est}}$, compared to using the second moment of the distance from the trap center. These results confirm the observation, that the position distribution at small distances carries limited information about the temperature.

Furthermore, we analyzed how the boson-impurity interaction strength affects thermometric sensitivity. In general, we found that the best precision is achieved with vanishingly small boson-impurity interaction strengths; however, this would prevent the impurity from thermalization. We observed that non-zero interaction strengths degrade thermometric performance, possibly due to the impurity deviating from thermal equilibrium, as suggested in Ref. [23]. Nevertheless, our numerical results suggest that for $0.9 \lesssim g_{BI}/g \lesssim 5$, the boson-impurity coupling has minimal impact on the impurity's thermometric performance. In the low-temperature limit, the impurity outperforms the estimation from a single atomic bath particle as a thermometer reaching the precision of $\Delta T \sim 1 - 1.5\%$ of $T_c$ for all analyzed $g_{IB}$.

In previous studies, reported in Ref. [23], the QFI sets a theoretical lower bound on temperature uncertainty, with estimates yielding a precision of 14% for 100 measurements, which translate to 4.4% for 1000 measurements, the latter being the benchmark employed in our work. Remarkably, our Monte Carlo finite-temperature results demonstrate a relative precision in the range of $\sim 4\%$ to 3%, positioning our approach as highly competitive with the QFI limit. Crucially, our method leverages experimentally accessible observables, namely, the spatial distribution of the impurity within the Bose gas, while achieving precision comparable with the fundamental bound set by the QFI.

In Ref. [23] it was indicated that estimating temperature based on the second moment of the impurity's position, $\langle |\mathbf{r}|^2 \rangle$, yields suboptimal uncertainties of 18% for 400 measurements, which translates to 11% for 1000 measurements used in our work. In contrast, our Monte Carlo simulations demonstrate that the thermometric approach outperforms these limits, achieving a relative uncertainty $\Delta T_{\text{est}}/T$ in the range of 3% to 5% in the weak coupling and 4% to 10% in the strong coupling limit.

Finally, Ref. [23] highlighted that increasing the probe-sample interaction strength does not enhance sensitivity, particularly in the regime of low trapping frequencies. In line with these findings, our results revealed a similar phenomenon, where the order of magnitude of the thermometric sensitivity remains largely unaffected by variations in the boson-impurity coupling strength, in our case $0 \leqslant g_{BI}/g \leqslant 5$. Remarkably, however, for large $g_{IB}/g$, the residual variations of $\Delta T_{\text{est}}$ with temperature are smaller compared to the case of small $g_{IB}/g$, where a noticeable linear trend in $T$ is observed. Such observations with our Monte Carlo results, reinforces the robustness of our method across a broad range of interaction strengths, further emphasizing its potential for practical applications in precision thermometry.

# Acknowledgements

**Funding information**  This work has been supported by the Spanish Ministerio de Universidades under the grant FPU No. FPU20/00013, by the Spanish Ministerio de Ciencia e Innovación (MCIN/AEI/10.13039/501100011033, grant PID2023-147469NB-C21), and by the Generalitat de Catalunya (grant 2021 SGR 01411). This research is part of the project No. 2021/43/P/ST2/02911 co-funded by the National Science Centre and the European Union Framework Programme for Research and Innovation Horizon 2020 under the Marie Skłodowska-Curie grant agreement No. 945339. For the purpose of Open Access, the author has applied a CC-BY public copyright licence to any Author Accepted Manuscript (AAM) version arising from this submission.

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
