# Peer review of "The Bose polaron as thermometer of a trapped Bose gas: a quantum Monte Carlo study"

_SciPost Physics_

## Round 1 · Referee Report · Anonymous (Referee 2) · 2025-5-11

Strengths

  1. The paper addresses an important problem in cold-atom physics
  2. The results presented in the paper are well-articulated.

Weaknesses

  1. The novelty of the presented study is unclear from the discussion in the text.
  2. An experimental protocol is not presented.

Report

The paper explores the possibility of measuring the temperature of a Bose gas using a mobile impurity. This is a timely topic because such measurements are challenging with standard techniques, such as time-of-flight, especially at temperatures much lower than the condensation temperature. The findings are clear and presented well. However, a number of issues should be addressed before this paper can be recommended for publication either in SciPost Physics or SciPost Physics Core.

  1. The logic of the manuscript seems to follow that of Ref. [23]. In particular, the paper confirms the main results of Ref. [23] and finds "that the actual thermometric performance is better when the Monte Carlo results are exploited". Note however that Ref. [23] considers a one-dimensional system, which complicates the direct comparison. From my point of view, the present study should rather compare to this paper: Physical Review Research 4, 023191 (2022).

  2. The manuscript does not explain when the present protocol outperforms other proposals in the field, e.g., Scientific Reports 4, 6436 (2014) Physical Review X 10, 011018 (2020), Eur. Phys. J. C (2023) 83:1022. In general, the first section does not provide a detailed introduction into the field of thermometry. At the very least, the first section should contain references that support the written statements. For example, the text "It is interesting to note, however, that beyond quantum simulation purposes, such as polaron physics and mediated interactions, investigations into exploiting impurities for technological applications, such as sensing devices, have only recently been undertaken." is unclear without references. The same is true for this sentence: "In a recent study, we have shown that the level of miscibility of impurities in a bosonic bath can be controlled by the bath’s temperature.", and a few other sentences in the introduction.

  3. The manuscript does not discuss a proof-of-concept experimental protocol to test the presented findings, making it difficult to use them directly in the existing laboratory setups. For example, the impurity and bosons have the same mass in the present study. In the corresponding experiments, the impurity is usually introduced by an RF pulse after the cooling process. The initial distribution of the impurities is then non-thermal, and some waiting time is required for the system to thermalize. If this time is too long then the suggested temperature measurement is probably not optimal.

  4. The manuscript repeatedly uses the term 'polaron,' which I believe is misleading in the present context. Indeed, the main results hold even without the impurity-boson interaction, indicating that the dressing of the impurity does not play a significant role. The paper also considers very strong interactions, for which the system is in the immiscible regime where the polaron concept is also inapplicable. It appears that the quasi-particle nature is not relevant to the present study.

  5. The manuscript focuses on 100 particles. This number is however too small for state-of-the-art three-dimensional experiments. The manuscript should comment on the applicability of the present results to actual experiments.

Recommendation

Ask for major revision

---

## Round 1 · Referee Report · Anonymous (Referee 3) · 2025-5-16

Strengths

  1. Interesting numerical results for the Bose polaron as thermometer of a Bose gas

  2. Interesting combination of Monte Carlo and estimation theory

Weaknesses

  1. Insufficient explanation of experimental protocols

  2. Reads more like a Scipost Core paper than a Scipost Physics

Report

The manuscript by Wasak et al. investigates the application of the Bose polaron as thermometer of a Bose gas. This problem has previously been proposed as a promising method for quantum nondemolition thermometers for Bose gases. The authors employ estimation theory using the path integral Monte Carlo method (PIMC) to analyze the impurity's sensitivity to temperature variations. They obtain the uncertainty of temperature measurements given by the Cramer-Rao lower bound and classical Fisher information of the k-th order statistics of the impurity position. Their findings show that the impurity located far from the centre of the harmonic trap captures more information about the temperature of the BEC than one positioned close to the centre.

Overall, the manuscript is well written and interesting. The authors present a clear and accurate description of estimation theory and derive improved results compared with those in Ref. [23], which are of particular interest for the field of quantum thermometry. However, I struggle to find the significance and novelty of the manuscript that satisfy the acceptance criteria of SciPost Physics. Instead, I would recommend the paper for publication in SciPost Physics Core.

Requested changes

  1. In the abstract, the authors state that the temperature range considered is -1.1 <= T/T_c <= 0.9. However, in the conclusion and in the figures, the investigated range appears to be 0.1 <= T/T_c <= 0.9. Could the authors clarify this apparent inconsistency?

  2. Several paragraphs in Section 3 are devoted to the formalism of the quantum Fisher information. However, the equations presented, particularly Eqs. (9)-(13), do not appear to be used in the later sections of the manuscript. I suggest moving this material to the appendix. If these equations do play a role in the main text, this connection should be clearly emphasized.

  3. It is mentioned that the position of the impurity can be measured non-destructively, however, the experimental protocol for such measurements is not clearly described. Could the authors provide more details on this?

  4. In Figs. 2, 3, and 5, the uncertainty of the temperature estimation is obtained by measuring a single boson from the cloud. Again, I would like the authors to comment on the experimental protocol for this measurement. Also, what is the role of the impurity in this measurement if a single boson can be measured non-destructively? How does this differ from the case without the impurity?

  5. In Fig. 2, it is mentioned that the standard deviation is scaled by a factor of 10. Based on the caption, does this mean that the actual uncertainty of the impurity's mean-squared distance is large and that the shaded regions corresponding to each line overlap more significantly? I would suggest that the authors rephrase their statement about the scaling to make the precise meaning clearer.

  6. The authors find that the best precision is achieved by measuring the impurity at vanishingly small boson-impurity interaction strengths. However, their calculations assume that the impurity thermalizes at any boson-impurity interaction strengths. This point is mentioned only in the conclusion. I suggest that the authors address this point earlier in the text.

Minor comments 1. The normalization in Section 4 and the equation for the likelihood function in Section 5 are not properly formatted.

  1. The color scheme used in Fig. 4 for different g_BI is not consistent with that in Figs. 2, 3, and 5. To facilitate comparison between the figures, could the authors use a consistent color scheme throughout?

  2. In the final paragraph of Section 8, it is stated that the weaker interacting case corresponds to 'blue' diamonds. Should this be 'red' instead?

Recommendation

Accept in alternative Journal (see Report)

---

## Editorial Decision

awaiting_resubmission